# Bilirubin Metabolism and Thyroid Cancer: Insights from ALBI and PALBI Indices

**DOI:** 10.3390/biom15071042

**Published:** 2025-07-18

**Authors:** Jong Won Shin, Jae Woong Sull, Nguyen Thien Minh, Sun Ha Jee

**Affiliations:** 1Department of Laboratory Medicine, Asan Medical Center, University of Ulsan College of Medicine, Ulsan, Seoul 05505, Republic of Korea; jongwon_shin@amc.seoul.kr; 2Department of Epidemiology and Health Promotion, Institute for Health Promotion, School of Public Health, Yonsei University, Seoul 03772, Republic of Korea; nguyenminh2301@gmail.com; 3Department of Biomedical Laboratory Science, College of Health Sciences, Eulji University, Seongnam 13135, Republic of Korea; jsull@eulji.ac.kr; 4Department of Epidemiology, Faculty of Public Health, University of Medicine and Pharmacy, Ho Chi Minh City 17000, Vietnam; 5Department of Transdisciplinary Healthcare Sciences, Graduate School of Transdisciplinary Health Science, Yonsei University, Seoul 06273, Republic of Korea

**Keywords:** bilirubin, albumin, thyroid neoplasms, oxidative stress, antioxidants

## Abstract

Background: This study evaluated the association between bilirubin subtypes (total, indirect, and direct bilirubin) and thyroid cancer risk, with a particular focus on stratified analyses using the ALBI (Albumin-Bilirubin) and PALBI (Platelet-Albumin-Bilirubin) indices by sex, smoking and drinking status, and age under 50 years. Methods: Data were obtained from 133,596 participants in the Korean Cancer Prevention Study-II (KCPS-II) cohort. During a mean follow-up period of 13.55 years, 2314 cases of thyroid cancer (ICD-10: C73) were identified. Serum bilirubin levels and ALBI and PALBI indices were analyzed using Cox proportional hazards regression models stratified by age, sex, smoking, and alcohol consumption status to estimate hazard ratios (HRs) and 95% confidence intervals (CIs). Results: In women, indirect bilirubin showed the strongest inverse association with thyroid cancer risk. ALBI and PALBI indices based on indirect bilirubin also demonstrated significant associations. A 1 standard deviation (SD) increase in indirect bilirubin was associated with a decreased risk of thyroid cancer (HR: 0.92, 95% CI: 0.84–0.99), and the ALBI index similarly showed an inverse association (HR: 0.92, 95% CI: 0.87–0.99). In contrast, the PALBI index was positively associated with thyroid cancer risk (HR: 1.11, 95% CI: 1.03–1.20). Among women who had never smoked, significant associations were observed for indirect bilirubin (HR: 0.91, 95% CI: 0.83–1.00), ALBI (HR: 0.93, 95% CI: 0.86–1.00), and PALBI (HR: 1.14, 95% CI: 1.05–1.23). In analyses stratified by alcohol consumption, the PALBI index was associated with increased thyroid cancer risk in non-drinkers, former drinkers, and ever drinkers, with respective risk increases of 15%, 18%, and 9%. Conclusions: In women, indirect bilirubin was significantly and inversely associated with thyroid cancer risk, and the ALBI and PALBI indices incorporating indirect bilirubin showed consistent results. These findings suggest that indirect bilirubin may play a critical role in the metabolic pathways underlying thyroid cancer in women.

## 1. Background

Thyroid cancer is one of the most common endocrine malignancies worldwide, with a rapidly increasing incidence in recent years [1,2]. In South Korea, it has the highest incidence rate among all cancers, posing a significant public health concern [3,4,5]. Well-established risk factors include radiation exposure, iodine intake, and genetic predisposition, but the role of metabolic and oxidative stress-related factors remains inadequately understood. Recent studies have highlighted the contribution of oxidative stress to thyroid carcinogenesis, leading to increasing interest in the potential protective role of antioxidants [6,7,8,9].

Bilirubin, a metabolic byproduct of heme catabolism, is a potent endogenous antioxidant that neutralizes reactive oxygen species (ROS) and reduces oxidative stress [10,11,12,13,14]. Previous research has primarily focused on total bilirubin and its association with cancer risk [15,16,17,18,19,20,21], while the relationship between bilirubin and thyroid cancer remains largely unexplored. In particular, no prior studies have examined the association between specific bilirubin subtypes (total, direct, and indirect bilirubin) and thyroid cancer risk, nor have they applied these subtypes to composite indices such as the ALBI (Albumin-Bilirubin) and PALBI (Platelet-Albumin-Bilirubin) indices.

Bilirubin subtypes differ in physiological function and may exert distinct effects on cancer risk [10,22,23]. Direct bilirubin, a conjugated and water-soluble form, is primarily excreted by the liver and plays a critical role in detoxification and hepatic function. In contrast, indirect bilirubin is an unconjugated, lipid-soluble form that serves as a major circulating antioxidant by participating in the NADH redox cycle to scavenge ROS. Given these biological distinctions, analyzing bilirubin subtypes separately may enhance the accuracy of cancer risk assessment.

In a previous study, we applied the ALBI index to examine the relationship between bilirubin subtypes and lung cancer risk and found that ALBI reflects not only the antioxidant role of bilirubin but also the non-enzymatic antioxidant properties of albumin, supporting its utility as a risk assessment index [24,25]. Building on this, the present study aimed to evaluate the associations of the ALBI and PALBI indices with bilirubin subtypes in relation to thyroid cancer risk. Although the PALBI index has been primarily used for prognostic assessment of liver function and hepatocellular carcinoma, its potential application in thyroid cancer risk assessment has not been investigated.

Furthermore, thyroid cancer is known to be highly sensitive to hormonal changes, and metabolic status varies by age and sex. In women, thyroid cancer is more frequently diagnosed at younger ages, particularly before menopause. In contrast, many other types of cancer are more common in older individuals [26]. In men, metabolic alterations related to age-associated declines in testosterone typically begin around the age of 50, potentially affecting metabolic homeostasis. Considering these sex- and age-specific metabolic features of thyroid cancer, we conducted stratified analyses among women under 50 years of age, representing a high-risk premenopausal group, and men under 50 years of age to assess risk in distinct metabolic contexts.

Additionally, bilirubin metabolism is closely linked to liver function, and both smoking and alcohol consumption are well-known contributors to oxidative stress and hepatic impairment. To further clarify the role of bilirubin and its indices in thyroid cancer development, we also conducted stratified analyses by smoking status and alcohol consumption. This approach may help elucidate how lifestyle-related oxidative stress modifies the antioxidant function of bilirubin and its potential utility in risk stratification for thyroid cancer.

## 2. Methods

This study utilized data from the Korean Cancer Prevention Study-II (KCPS-II), which included 153,971 adults aged 20 years or older who visited 18 health examination centers across South Korea between 2004 and 2013. All participants provided written informed consent for the use of their health examination and survey data for research purposes [27]. The baseline of this study was defined as the time point when participants underwent health examinations and blood sampling. From this point forward, participants were prospectively followed to track long-term health outcomes, including cancer incidence. Individuals who had been diagnosed with cancer or identified as cancer survivors before enrolling in the cohort were excluded from the study. Participants with missing data on smoking or alcohol consumption were also excluded. Lastly, individuals with missing values for key biomarkers, including bilirubin, albumin, platelet count, and other biochemical indicators used for confounding adjustment, were further excluded. As a result, the final analytic sample consisted of 133,596 participants. Final analyses were conducted separately for the total population and the subgroup under 50 years of age. The total study population included 133,596 adults aged 20 years or older, while the subgroup under 50 years comprised 107,851 individuals from the total sample. This study was approved by the Institutional Review Board (IRB) of Yonsei University Severance Hospital, with approval number 4-2011-0277.

### 2.1. Bilirubin Measurement

Serum bilirubin concentrations were assessed using an automated clinical chemistry analyzer (AU5800; Beckman Coulter, Seoul, Republic of Korea). Total bilirubin was quantified based on its reaction with 3,5-dichlorophenyldiazonium tetrafluoroborate (DPD), a stabilized diazonium compound, forming an azobilirubin complex. To facilitate this reaction, reagents including caffeine and surfactants were added. The resulting azobilirubin was detected by measuring absorbance at dual wavelengths (570/660 nm), which correlates linearly with the concentration of bilirubin in the sample. A serum blank was also processed to account for potential background interference from endogenous substances. The assay demonstrated a within-run coefficient of variation (CV) below 3% or a standard deviation (SD) of ≤0.07, while total assay precision remained under a CV of 5% or an SD of ≤0.10. For direct bilirubin, a modified version of the classical method developed by Van den Bergh and Mueller was used [28]. In this approach, conjugated bilirubin reacts with DPD under acidic conditions to form azobilirubin, and the absorbance is measured at 540/600 nm. The intensity of the color produced is proportional to the concentration of direct bilirubin in the serum. The within-run CV for this assay was less than 7%, and the SD did not exceed 0.07. The total CV was below 8%, with an SD of ≤0.21. Indirect bilirubin levels were not directly measured but calculated by subtracting the direct bilirubin concentration from the total bilirubin value.

### 2.2. Cancer Case Ascertainment

Cancer incidence among the study participants was determined with nearly 100% accuracy by annually linking their resident registration numbers to the National Cancer Center (NCC) Registry [27]. In South Korea, all hospitals are mandated by the Cancer Control Act to report cancer cases to the NCC. Thyroid cancer cases were classified according to the 10th revision of the International Classification of Diseases (ICD-10) as C73. The average follow-up period for the entire cohort was 13.55 years, during which a total of 2314 thyroid cancer cases were identified. This number reflects the analysis conducted on the total population. The median follow-up duration was 14.0 years (IQR: 13.4–14.6). Additionally, a subgroup analysis was conducted on individuals aged under 50 years. In this group, 1860 thyroid cancer cases were identified, with a median follow-up duration of 14.0 years (IQR: 13.4–14.4).

### 2.3. Statistical Analysis

Descriptive statistics were used to summarize the general characteristics of the study population. Bilirubin subtypes (total, direct, and indirect bilirubin), as well as the ALBI and PALBI indices, were categorized into quartiles to evaluate their associations with thyroid cancer risk. A trend analysis was conducted to assess differences in thyroid cancer incidence across quartiles. Additionally, the effects of a one-standard deviation (1-SD) increase in each bilirubin subtype and the ALBI/PALBI indices on thyroid cancer risk were assessed. Cox proportional hazards regression models were used to estimate hazard ratios (HRs) and 95% confidence intervals (CIs) for the associations between bilirubin-related markers and thyroid cancer incidence. These models were adjusted for potential confounders including age, smoking history, alcohol consumption, body mass index (BMI), aspartate aminotransferase (GOT), alanine aminotransferase (GPT), blood urea nitrogen (BUN), and family history of thyroid cancer. A complete-case analysis was performed, excluding participants with missing covariate data. The overall proportion of missing data was very low, with most variables having less than 1% missingness, minimizing potential bias. The ALBI index was calculated using the following formula: ALBI = (log_10_ bilirubin [μmol/L] × 0.66) + (albumin [g/L] × −0.085) [29]. For this calculation, bilirubin values were converted from mg/dL to μmol/L. The PALBI index was calculated using the following formula: PALBI = log_10_(platelet count [×10^9^/L]) − 2.02 × serum albumin [g/dL] − 0.37 × log_10_(serum bilirubin [mg/dL]) [30]. To evaluate the role of each bilirubin subtype, total, direct, and indirect bilirubin values were separately applied to the ALBI and PALBI formulas, generating subtype-specific indices and corresponding hazard ratios. Stratified analyses were conducted according to smoking status (never, former, current, ever) and alcohol consumption (never, former, current, ever) to evaluate potential effect modification. Sensitivity analyses excluding liver function-related biomarkers (e.g., GOT, GPT, BUN) were also performed to assess the robustness of the associations. Additionally, to account for age-related hormonal changes, such as menopause, separate stratified analyses were performed among participants younger than 50 years in both men and women. All biochemical measurements were conducted in certified laboratories that followed standardized internal and external quality control protocols designated by the Korean Association of Laboratory Quality Control. The inter-laboratory correlation coefficients for the measured biomarkers ranged from 0.96 to 0.99, indicating high accuracy and consistency across all testing centers [27]. All statistical analyses were performed using SAS version 9.4 (SAS Institute, Cary, NC, USA).

## 3. Results

Baseline characteristics, including age, body mass index, bilirubin, albumin, platelet count, alcohol consumption, and smoking status, are presented in Table 1. In the quartile analysis (Q1–Q4), indirect bilirubin in women showed statistically significant inverse associations with thyroid cancer in Q2 (HR: 0.84, 95% CI: 0.71–0.99) and Q4 (HR: 0.79, 95% CI: 0.65–0.97), with a significant trend (*p* for trend = 0.0221). Similarly, the ALBI index based on indirect bilirubin showed inverse associations in Q2 (HR: 0.84, 95% CI: 0.71–0.98) and Q4 (HR: 0.79, 95% CI: 0.65–0.97) (*p* for trend = 0.0209). In men, the ALBI index incorporating indirect bilirubin showed a significant inverse association in Q2 (HR: 0.80, 95% CI: 0.64–0.99) (Table 2).

Among women, indirect bilirubin was inversely associated with thyroid cancer risk, with an HR of 0.92 (95% CI: 0.84–0.99) per 1 SD increase. The ALBI index incorporating indirect bilirubin showed a similar inverse association (HR: 0.92, 95% CI: 0.87–0.99), while the PALBI index showed a positive association (HR: 1.11, 95% CI: 1.03–1.20) (Figure 1).

Among women under the age of 50, total bilirubin was inversely associated with thyroid cancer risk (HR: 0.92, 95% CI: 0.85–0.99). The ALBI index based on total bilirubin showed a similar inverse association (HR: 0.93, 95% CI: 0.87–0.99), whereas the PALBI index showed a positive association (HR: 1.07, 95% CI: 1.00–1.15). Indirect bilirubin was also inversely associated with risk (HR: 0.91, 95% CI: 0.83–1.00), and the ALBI index based on it showed an HR of 0.92 (95% CI: 0.86–0.99). The PALBI index, in contrast, showed a positive association (HR: 1.10, 95% CI: 1.01–1.19) (Appendix A).

In men, no statistically significant associations were observed for any bilirubin subtypes (Figure 1). However, among men under 50 years of age, the PALBI index incorporating indirect bilirubin showed a borderline significant positive association (HR: 1.09, 95% CI: 1.00–1.18) (Appendix A).

In the stratified analysis by smoking status, a positive association was observed in past-smoking men, where the PALBI index based on indirect bilirubin was associated with increased thyroid cancer risk (HR: 1.18, 95% CI: 1.03–1.35) (Appendix A). In contrast, statistically significant associations in women were observed only among never-smokers. Among never-smoking women, indirect bilirubin and the ALBI index based on it were inversely associated with thyroid cancer (HR: 0.91, 95% CI: 0.83–1.00 and HR: 0.93, 95% CI: 0.86–1.00, respectively). The PALBI index showed positive associations when applied with each bilirubin subtype: total (HR: 1.12, 95% CI: 1.03–1.22), indirect (HR: 1.14, 95% CI: 1.05–1.23), and direct bilirubin (HR: 1.10, 95% CI: 1.02–1.19) (Appendix A).

In the stratified analysis by alcohol consumption, no significant associations were found in men (Appendix A). In women, however, the PALBI index based on indirect bilirubin showed positive associations with thyroid cancer among never drinkers (HR: 1.15, 95% CI: 1.00–1.32), former drinkers (HR: 1.18, 95% CI: 1.00–1.39), and ever drinkers (HR: 1.09, 95% CI: 1.00–1.20) (Appendix A).

In sensitivity analyses excluding liver-related biomarkers such as GOT, GGT, and BUN, the findings remained consistent. In women, both indirect bilirubin (HR: 0.92, 95% CI: 0.85–1.00) and the ALBI index (HR: 0.93, 95% CI: 0.87–1.00) showed inverse associations, while the PALBI index showed a positive association (HR: 1.09, 95% CI: 1.01–1.18) (Appendix A).

## 4. Discussion

This study is the first to evaluate the association between bilirubin subtypes (total, indirect, and direct bilirubin) and thyroid cancer risk using the ALBI (Albumin-Bilirubin) and PALBI (Platelet-Albumin-Bilirubin) indices. Among women, indirect bilirubin and the indices incorporating it (ALBI and PALBI) showed significant associations with thyroid cancer risk. These associations remained consistent across the overall population, the subgroup under age 50, and in sensitivity analyses excluding liver-related biomarkers (Figure 1 and Appendix A). In stratified analyses by smoking status, significant associations were most prominent in non-smoking women (Appendix A), while in drinking status analysis, the PALBI index incorporating indirect bilirubin showed a significant positive association with thyroid cancer risk (Appendix A). In men, no significant associations were observed across age groups, but among those under age 50, the PALBI index exhibited a borderline positive association with thyroid cancer risk. These findings suggest potential sex-based differences in the role of bilirubin metabolism in thyroid cancer development, with a particularly notable inverse association between indirect bilirubin and thyroid cancer risk in women. Importantly, the KCPS-II cohort used in this study consists of a relatively young population (mean age ~40 years) (Table 1), providing valuable insights into the metabolic risk factors associated with early-onset thyroid cancer. Recent studies have primarily applied the ALBI and PALBI indices to assess prognosis in cancer patients, such as those with breast, colorectal, or lung cancer [31,32,33,34]. While such studies have offered important insights into post-diagnostic pathological conditions, they are often subject to substantial confounding due to study design limitations. In contrast, the current study applied these indices to a healthy general population to assess thyroid cancer risk, distinguishing it from previous research. The results demonstrated that bilirubin subtypes included in the ALBI and PALBI indices exhibit contrasting associations with thyroid cancer according to sex and age, emphasizing the need for further physiological studies to elucidate the underlying mechanisms of thyroid cancer development.

### Redox Metabolism and Thyroid Cancer

Unconjugated bilirubin (indirect bilirubin) binds to albumin in the bloodstream and is transported to the liver, where it acts as a potent antioxidant by neutralizing reactive oxygen species (ROS) through the NADH redox process. NADH is a key enzymatic system that regulates redox reactions in the body, and indirect bilirubin helps modulate the NADH/NAD^+^ ratio to suppress and eliminate ROS. ROS can initiate inflammatory responses, cause DNA damage, and lead to oncogenic mutations such as those in BRAF and RAS genes, thereby increasing the aggressiveness and progression of thyroid cancer [6,7,8,9,35,36]. In this study, the ALBI index incorporating indirect bilirubin showed a strong inverse association with thyroid cancer, suggesting that the ROS-neutralizing effects of indirect bilirubin may play a protective role in thyroid cancer prevention.

Albumin not only serves as a carrier for indirect bilirubin but also functions as a non-enzymatic antioxidant by binding to metal ions and scavenging ROS. It enhances the antioxidant activity of indirect bilirubin through complex formation, thereby contributing to cellular protection against oxidative stress [37]. The interaction between indirect bilirubin and albumin may help shield thyroid cells from oxidative damage and contribute to cancer prevention. In addition, indirect bilirubin is likely to interact with tyrosine kinase receptor pathways, potentially regulating MAPK and PI3K/Akt signaling in thyroid cells [36]. These mechanisms may help stabilize signaling pathways and suppress tumorigenesis by reducing oxidative stress.

Thyroid hormones also regulate the hepatic metabolism and excretion of bilirubin [38,39]. In hypothyroidism, decreased activity of UDP-glucuronosyltransferase (UGT) slows the conjugation of bilirubin in the liver, resulting in increased levels of indirect bilirubin. In contrast, in hyperthyroidism, enhanced UGT activity accelerates the conversion of indirect to direct bilirubin and increases bile excretion. These metabolic changes can intensify oxidative stress in hepatocytes and lead to hepatic dysfunction and metabolic imbalance [40].

Albumin also binds to thyroid hormones such as thyroxine (T4) and triiodothyronine (T3), facilitating their transport to target tissues [41]. In cases of albumin deficiency, impaired hormone binding and transport can lead to imbalances in free thyroid hormone levels, elevated oxidative stress, and thyroid dysfunction [42]. The liver acts as a central organ for thyroid hormone metabolism, protein synthesis, and bile excretion, and plays a pivotal role in the metabolic interactions between the thyroid and bilirubin [38,39]. Elevated levels of oxidative stress biomarkers, such as malondialdehyde (MDA) and 8-hydroxy-2′-deoxyguanosine (8-OHdG), have been reported in patients with thyroid cancer, supporting the role of ROS in thyroid cancer development and progression [41]. The ALBI index reflects these integrated metabolic and oxidative processes and may serve as a useful tool for understanding the mechanisms underlying reduced thyroid cancer risk.

The PALBI index integrates platelet count, albumin, and bilirubin and may represent an expanded marker beyond liver function. It potentially reflects metabolic mechanisms involved in tumor development and progression. In this study, the PALBI index was positively associated with thyroid cancer risk, which aligns with prior evidence that platelets may contribute to tumor growth and metastasis. Traditionally recognized for their role in coagulation and hemostasis, platelets have also emerged as key regulators within the tumor microenvironment. Activated platelets can interact with cancer cells, enhancing their proliferation, motility, and invasiveness. Studies have shown that patients with thyroid cancer exhibit significantly increased mean platelet volume (MPV) and platelet distribution width (PDW) compared to healthy individuals, suggesting that platelet activation may facilitate cancer progression [42,43].

Furthermore, certain chemokines secreted by platelets, particularly CCL3, can activate the matrix metalloproteinase-1 (MMP-1) pathway, promoting tumor cell invasion [44]. This provides a physiological basis for the observed positive association between higher platelet counts and increased thyroid cancer risk as reflected by the PALBI index. Platelet activation also promotes ROS production via the NOX4 (NADPH oxidase 4) pathway. This, in turn, upregulates oncogenes such as BRAF and RAS, activates the MAPK/ERK signaling pathway, and facilitates tumor progression [36]. In thyroid cancer, elevated NOX4 expression has been linked to increased oxidative stress, DNA damage, and genomic instability, which further contribute to tumor aggressiveness and progression [6,7,8,9,36].

Additionally, thyroid hormones have been shown to influence platelet activation and tumor progression. In hyperthyroid states, elevated platelet counts may amplify inflammation within the tumor microenvironment and promote the motility and invasiveness of cancer cells [42]. Thyroid hormones also stimulate angiogenesis via the integrin αvβ3 pathway, increasing oxygen and nutrient supply to tumors and accelerating their growth [45]. In conclusion, platelets, as a key component of the PALBI index, may function not only as indicators of liver function but also as significant physiological mediators in the tumor microenvironment, influencing cancer cell invasiveness and metastatic potential.

The findings of this study suggest that serum indirect bilirubin may serve as a potential biomarker for thyroid cancer risk, particularly in women. Given its inverse association with thyroid cancer, low levels of indirect bilirubin may reflect impaired antioxidant defense and increased vulnerability to oxidative stress-related carcinogenesis in thyroid tissue. In clinical settings, if a patient, especially a younger woman, presents with persistently low serum indirect bilirubin levels, this may warrant further evaluation. Although bilirubin alone is not sufficient for diagnosis, its role as an accessible, cost-effective, and routinely measured blood parameter could be integrated into risk stratification models. In such cases, clinicians may consider additional diagnostic workup, including thyroid ultrasound, serum thyroid-stimulating hormone (TSH) testing, or other imaging studies, particularly in individuals with additional risk factors such as family history, radiation exposure, or nodular thyroid disease. Incorporating bilirubin levels into preventive screening strategies could help identify subpopulations that may benefit from closer surveillance. Furthermore, these findings highlight the need for future research to explore bilirubin metabolism and its modulation as a potential avenue for thyroid cancer prevention.

## 5. Conclusions

This study is the first cohort-based investigation to evaluate the association between bilirubin subtypes (total, indirect, and direct bilirubin) and thyroid cancer risk using the ALBI (Albumin-Bilirubin) and PALBI (Platelet-Albumin-Bilirubin) indices in a general population. Indirect bilirubin, known for its strong antioxidant properties, showed an inverse association with thyroid cancer, and the ALBI index incorporating indirect bilirubin consistently reflected reduced risk. In contrast, the PALBI index that includes indirect bilirubin showed a positive association with thyroid cancer risk, suggesting divergent roles of these indices. These findings imply that ALBI and PALBI indices may reflect not only liver function but also broader physiological mechanisms, such as oxidative stress and the tumor microenvironment. Specifically, the ALBI index may capture the antioxidative interaction between indirect bilirubin and albumin, which could play a protective role against thyroid cancer development. Meanwhile, the PALBI index may indicate platelet-related pathways that promote tumor progression. Although these indices are not yet applicable as early diagnostic tools, our findings provide a foundation for future research into the metabolic mechanisms of thyroid cancer, potential sex-specific differences, and strategies for risk stratification based on bilirubin metabolism.

## 6. Limitations

Several limitations should be considered when interpreting the findings of this study.

First, as a cohort study, this research examined the associations between bilirubin subtypes, the ALBI/PALBI indices, and thyroid cancer risk. However, it did not establish definitive causal relationships. Experimental or interventional studies are needed to clarify these associations and the underlying biological mechanisms. Second, the study population consisted of Korean and Asian individuals. This may limit the generalizability of the findings to populations with different genetic and environmental backgrounds. Further validation in Western and other diverse populations is warranted. Third, environmental and lifestyle factors that may influence thyroid cancer development, such as dietary patterns, physical activity, and socioeconomic status, were not fully accounted for in the analysis. In addition, important potential confounders including socioeconomic variables, hormonal factors, menopausal status, family history, and medication use were only partially adjusted for. Therefore, there remains the possibility of residual confounding. Future studies should incorporate these variables more comprehensively. Fourth, although the study included a relatively long follow-up period (mean 13.55 years), thyroid cancer often has a long latency period. Longer-term observational studies may be necessary to more accurately evaluate the long-term effects of bilirubin subtypes and ALBI/PALBI indices on thyroid cancer risk. Fifth, bilirubin levels and the ALBI/PALBI indices were measured at a single time point. As a result, this study did not capture temporal changes in these biomarkers, which could be important for understanding dynamic associations with thyroid cancer risk. Future research should consider repeated measurements to assess longitudinal patterns. Finally, this study focused exclusively on thyroid cancer and did not investigate associations of bilirubin subtypes or ALBI/PALBI indices with other cancer types. Additional research is needed to explore whether these findings extend to other malignancies and to assess the broader clinical utility of these indices.

## Figures and Tables

**Figure 1 biomolecules-15-01042-f001:**
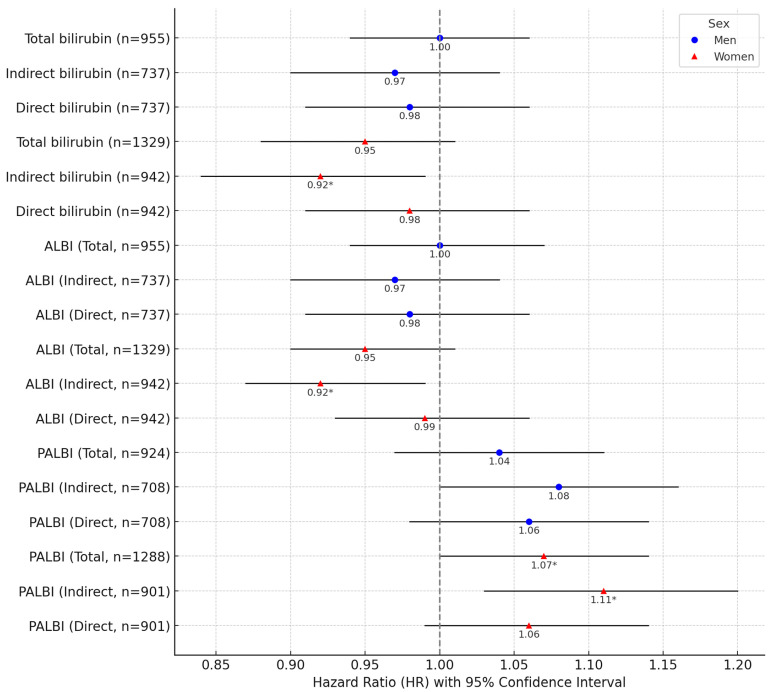
Association of thyroid cancer risk with a 1 SD increase in bilirubin subtypes, ALBI, and PALBI in men and women of all ages. Note: Error bars represent 95% confidence intervals. * *p* < 0.05.

**Table 1 biomolecules-15-01042-t001:** Baseline characteristics of Korean cancer prevention study-II participants *.

Characteristics	Men (n = 83,371)	Men (n = 66,875, Age < 50)	Women (n = 50,225)	Women (n = 40,976, Age < 50)
Age, y	41.6 (9.5)	37.93 (6.00)	39.7 (10.7)	35.81 (7.07)
Body mass index ^†^	24.4 (2.9)	24.42 (2.97)	22.0 (3.0)	21.53 (2.86)
Serum bilirubin, mg/dL				
Total	0.95 (0.38)	0.95 (0.38)	0.75 (0.30)	0.76 (0.31)
Indirect	0.59 (0.26)	0.59 (0.26)	0.47 (0.21)	0.47 (0.21)
Direct	0.36 (0.14)	0.36 (0.14)	0.29 (0.12)	0.29 (0.12)
Albumin, g/dL	4.58 (0.25)	4.59 (0.24)	4.45 (0.25)	4.45 (0.25)
platelet, 10^3^/µL	244.69 (48.91)	246.82 (48.30)	254.36 (51.28)	254.53 (51.22)
Alcohol drinking, g/d	22.67 (29.45)	22.40 (27.91)	5.98 (13.77)	6.65 (14.98)
Smoking status, %				
Never	22.7	22.8	89.4	88.7
Previous	32.7	29.5	6.4	6.7
Current	44.6	47.7	4.2	4.6
Any alcohol use, %				
Never	5.8	4.80	30.8	24.4
Previous	7.9	7.6	16.5	17.8
Current	86.3	87.5	52.7	57.9

* Data are expressed as mean (SD) unless otherwise indicated. Participants with any of the following features at study entry were excluded: missing data on serum bilirubin level, existing cancer, and missing data on smoking status. ^†^ Body mass index was calculated as weight in kilograms divided by the square of height in meters.

**Table 2 biomolecules-15-01042-t002:** Association of bilirubin subtypes, ALBI, and PALBI with thyroid cancer risk: Quartile and trend analysis by sex.

	Bilirubin Subtypes	Case	Q1 HR (95% CI) ^§^	Q2 HR (95% CI) ^§^	Q3 HR (95% CI) ^§^	Q4 HR (95% CI) ^§^	*p* Value for Trend
Men	Total Bilirubin	955	1	1.03 (0.86–1.25)	0.94 (0.77–1.14)	1.00 (0.83–1.20)	0.7811
	Indirect Bilirubin	737	1	1.00 (0.79–1.26)	0.87 (0.69–1.08)	0.91 (0.73–1.13)	0.2385
	Direct Bilirubin	737	1	1.03 (0.84–1.25)	0.96 (0.77–1.20)	0.92 (0.74–1.13)	0.3103
Women	Total Bilirubin	1329	1	0.93 (0.82–1.06)	0.93 (0.79–1.09)	0.89 (0.75–1.06)	0.1560
	Indirect Bilirubin	942	1	0.84 (0.71–0.99)	0.88 (0.74–1.04)	0.79 (0.65–0.97)	0.0221
	Direct Bilirubin	942	1	0.97 (0.83–1.13)	1.02 (0.83–1.24)	1.02 (0.82–1.26)	0.8734
	ALBI Index						
Men	Total Bilirubin	955	1	1.10 (0.90–1.33)	0.94 (0.77–1.14)	1.03 (0.85–1.24)	0.7803
	Indirect Bilirubin	737	1	0.80 (0.64–0.99)	0.89 (0.72–1.10)	0.84 (0.69–1.04)	0.3042
	Direct Bilirubin	737	1	1.03 (0.83–1.28)	0.98 (0.79–1.21)	0.90 (0.73–1.11)	0.2237
Women	Total Bilirubin	1329	1	0.92 (0.80–1.05)	0.87 (0.75–1.01)	0.89 (0.75–1.05)	0.0639
	Indirect Bilirubin	942	1	0.84 (0.71–0.98)	0.87 (0.73–1.04)	0.79 (0.65–0.97)	0.0209
	Direct Bilirubin	942	1	1.02 (0.87–1.21)	0.95 (0.80–1.13)	1.01 (0.83–1.22)	0.8047
	PALBI Index						
Men	Total Bilirubin	924	1	0.93 (0.79–1.11)	0.99 (0.83–1.18)	1.04 (0.86–1.26)	0.6764
	Indirect Bilirubin	708	1	0.94 (0.77–1.15)	1.10 (0.90–1.34)	1.09 (0.88–1.35)	0.2563
	Direct Bilirubin	708	1	1.10 (0.90–1.34)	1.15 (0.93–1.41)	1.16 (0.93–1.45)	0.1451
Women	Total Bilirubin	1288	1	0.98 (0.81–1.20)	1.07 (0.89–1.29)	1.12 (0.93–1.33)	0.0967
	Indirect Bilirubin	901	1	0.92 (0.73–1.17)	1.15 (0.92–1.42)	1.18 (0.96–1.45)	0.0185
	Direct Bilirubin	901	1	0.97 (0.78–1.21)	1.04 (0.85–1.28)	1.10 (0.90–1.34)	0.2172

Abbreviations: CI, confidence interval; HR, hazard ratio. ^§^ The Cox proportional hazards model was adjusted for age, smoking status, alcohol use, body mass index, GOT, GGT, BUN, and family history.

## Data Availability

The datasets, including summary statistics and R code generated during the current study, are available from the corresponding author upon reasonable request. Due to privacy and ethical restrictions, raw data are not publicly available.

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
