# Peer review of "Bilirubin Metabolism and Thyroid Cancer: Insights from ALBI and PALBI Indices"

_biomolecules, 2025, doi:10.3390/biom15071042_

Round 1
Reviewer 1 Report
Comments and Suggestions for Authors
Shin et al. examined the association of bilirubin with thyroid cancer risk using a large Korean cohort study. They evaluated the subgroups of bilirubin (total, direct, and indirect), as well as other indices, such as ALBI and PALBI. However, there are some concerns that need to be addressed as follows.
- Was the same machine used to measure bilirubin levels across 18 centers?
- The measurements of other covariates, such as smoking, alcohol, BMI, GOT, GPT, albumin, and platelet count, should be mentioned in the Methods section. How did the authors deal with missing information on these covariates?
- Line 175: please revise In women, indirect bilirubin was negatively associated with thyroid cancer risk (HR 0.921, 95% CI: 0.85-0.998 per 1 SD increase)
- Data presentation.
- For the figure, suggest showing both men and women in all age groups to show the differences between sex groups. I think the results were quite consistent in the subgroup stratified by 50 years. Suggest moving these results to the supplement. Since the majority of women in the current study are under 50 years old, this finding is intuitive.
- I just wanted to make sure. Are all women under the age of 50 premenopausal? Please clarify. Also, how did the authors get the information on menopausal status?
- Suggest showing Table 2 first. It would be worth showing first whether the association is linear or nonlinear. Once the relationship is confirmed as linear, it would be worth showing the association of continuous bilirubin levels (1SD) with thyroid cancer risk since the current data has enough cases.
- Table 2: Delete Median follow-up time, which made the table very messy. Show cases by each quartile and the range of levels in each quartile.
- Have the authors tried to conduct other subgroup analyses, such as smoking status, alcohol status, and liver enzyme levels? These are strong risk factors for cancer risk (maybe not specifically in thyroid cancer, though). Worth checking whether these factors can be confounded or mediators.
- Line 229: What do you mean “these trends remained consistent even after adjusting for premenopausal age.”? Did the authors additionally adjust for age in the cox model? Please clarify.
- In the Discussion, I expected that the first paragraph would describe the general summary of findings, but it suddenly mentioned mechanisms and then findings from previous studies. Recommend having one theme in each paragraph. It is hard to follow the logic.
- Line 333: “While the direction of association for ALBI and PALBI indices was similar in both men and women, the statistical significance varied. This discrepancy is likely due to differences in statistical power stemming from the sex-specific distribution of thyroid cancer” Have the authors tested the interactions between sex? Authors mentioned that sex differences were seen, but a non-significant association in men might be due to weak statistical power, which does not match.
- What are the clinical implications of these findings? How can we use bilirubin as a marker of thyroid cancer? If the bilirubin level is too low, what are the next steps to take to identify cancer?
Reviewer 2 Report
Comments and Suggestions for Authors
The authors presented a manuscript entitled “Bilirubin Metabolism and Thyroid Cancer: Insights from ALBI and PALBI Indices” The article theme is good, however, I feel the article can be suitable for publishing after a major revision to match the standards of the MDPI Biomolecules. I suggest a Major revision.
My suggestions
1. Were the values of bilirubin log-transformed or normalized prior to calculation?
2. What type of bilirubin (total, direct, or indirect) was utilized in these indices? Did the authors try calculations across different types, and did they report the outcomes?
3. On what basis would one prioritize indirect bilirubin over direct or total bilirubin? Were correlation coefficients across these subtypes tested to prevent multicollinearity?
4. Might the observed associations be partly accounted for by covert hemolytic states or liver impairment affecting levels of indirect bilirubin?
5. Were proportional hazards assumptions checked for the Cox models employed? If they were, how were these checked and validated?
6. Was sensitivity analysis or stratified analysis of any kind performed to investigate robustness of association?
7. While hazard ratios report statistical significance, were any of the effect sizes clinically significant? For instance, what is the absolute reduction in risk at the highest vs. lowest quartile of bilirubin or ALBI/PALBI?
Round 2
Reviewer 2 Report
Comments and Suggestions for Authors
The article can be accepted in its present form.